# Exploring patient-, provider-, and health facility-level determinants of blood pressure among patients with hypertension: A multicenter study in Ghana

Samuel Byiringiro[1]*, Thomas Hinneh[1], Yvonne Commodore-Mensah[1,2], Jill Masteller[2], Fred Stephen Sarfo[3,4], Nancy Perrin[1], Shadrack Assibey[4], Cheryl R. Himmelfarb[1,2,5]

1 Johns Hopkins University, School of Nursing, Baltimore, Maryland, United States of America, 2 Johns Hopkins University, Bloomberg School of Public Health, Baltimore, Maryland, United States of America, 3 Department of Medicine, Kwame Nkrumah University of Science & Technology, Kumasi, Ashanti Region, Ghana, 4 Komfo Anokye Teaching Hospital, Kumasi, Ashanti Region, Ghana, 5 Johns Hopkins University, School of Medicine, Baltimore, Maryland, United States of America

* sbyirin1@jhu.edu

**Data Availability Statement:** The data analyzed is available in an online repository: https://www.openicpsr.org/openicpsr/project/206321/version/

## Abstract

Optimal blood pressure (BP) control is essential in averting cardiovascular disease and associated complications, yet multiple factors influence the achievement of BP targets. We explored patient-, provider-, and health facility-level factors of systolic and diastolic BP and controlled BP status among patients with hypertension in Ghana. Using a cross-sectional design, we recruited 15 health facilities, and from each facility, we recruited four healthcare providers involved in managing hypertension and 15 patients diagnosed with hypertension. The primary outcome of interest was systolic and diastolic BP; the secondary outcome was BP control (<140/90 mmHg) in compliance with Ghana's national standard treatment guidelines. We used mixed-effects regression models to explore the patient- and facility-level predictors of the outcomes. Two hundred twenty-four patients and 67 healthcare providers were sampled across 15 health facilities. The mean (SD) age of providers and patients was 32 (7) and 61 (13) years, respectively. Most (182 [81%]) of the patient participants were female, and almost half (109 [49%]) had controlled BP. At the patient level, traveling for 30 minutes to one hour to the health facility was associated with higher diastolic BP (Coeff.:3.75, 95% CI: 0.12, 7.38) and lower odds of BP control (OR: 0.51, 95% CI: 0.28, 0.92) compared to traveling for less than 30 minutes. Receiving hypertension care at government health facilities than at private health facilities was associated with lower systolic BP (Coeff.: -13.89; 95% CI: -23.99, -3.79). A higher patient-to-physician or physician-assistant ratio was associated with elevated systolic BP (Coeff.: 21.34; 95% CI: 8.94, 33.74) and lower odds of controlled BP (OR: 0.19, 95% CI: 0.05, 0.72). Along with addressing the patient-level factors influencing BP outcomes in Ghana, there is a need for public health and policy interventions addressing the inaccessibility of hypertension services, the shortage of clinical care providers, and the underperformance of private health facilities.

V1/view. Please reach out to the corresponding author for any questions.

**Funding:** The present study received funding from the Discovery and Innovation Fund by the Johns Hopkins School of Nursing, the Johns Hopkins Provosts' Dissertation Award, and the Global Health Established Field Placement Travel Grant offered by the Johns Hopkins Bloomberg School of Public Health. In implementing this study, SB received additional funding from the Joana and Bill Conway Scholarship (2019-2022). During the manuscript write-up, SB was funded by the American Heart Association (23DIVSUP1058025). The funders had no role in study design, data collection and analysis, publication decision, or manuscript preparation.

**Competing interests:** The authors have declared that no competing interests exist.

# Background

A quarter of Sub-Saharan Africa's (SSA) adult population has hypertension, a major risk factor for cardiovascular diseases, the region's leading cause of death [1,2]. In Ghana, approximately 27% of the adult population has hypertension [3]. Additionally, there is a low prevalence of hypertension awareness, diagnosis, and treatment, which is one of the reasons why only 6% of all people with hypertension have controlled BP [3]. Countries in SSA are predominantly classified by the World Bank as low-income and usually have resource-constrained health systems that cannot respond to the projected further rise in hypertension and other non-communicable diseases (NCDs) [4]. Inadequate financing of the health systems contributes to the shortcomings across key health system components, including the lack of essential equipment, insufficient training of staff, and inconsistent stocking of health facility pharmacies and other clinical consumables [5,6]. These challenges further limit the population's awareness of the dangers of major NCDs, such as hypertension, hence leaving most patients undiagnosed and, therefore, without access to treatment [7–9]. Multi-level interventions will be necessary to respond to SSA's rising burden of hypertension.

While hypertension is a chronic condition with life-threatening complications, these can be averted by controlling BP using lifestyle modification practices, antihypertensive therapy, or both [10]. Multiple guidelines with different thresholds for defining hypertension and initiating patients on treatment exist [10–13]. In its 2021 roadmap for hypertension management, the World Heart Federation and many countries in SSA, including Ghana, recommend treatment of hypertension to the target of $\leq$ 140/90 mmHg [13,14]. Despite variabilities in BP treatment targets, evidence shows that even among patients with uncontrolled hypertension, a 5-mmHg reduction in systolic or diastolic BP is associated with better health outcomes through the reduction of composite cardiovascular event risk [15,16]. Therefore, it is beneficial to explore the factors that influence systolic and diastolic BP reduction and the overall achievement of hypertension control.

Apart from health system inadequacies, additional factors influence hypertension care and outcomes at the individual patient level, including age, sex, socioeconomic status, education, employment, family history of hypertension, obesity, and lifestyles such as smoking, high salt consumption, excessive caloric intake, alcoholism, and the lack of physical exercise [17]. There are very interlinked influences among the various risk levels, and all contribute to better or poor hypertension outcomes [18]. Despite the multi-level and multifaceted nature of factors contributing to hypertension outcomes, there is a lack of understanding of the cross-influence of those multi-level factors. The present study aims to explore the health facility readiness for hypertension management and assess the patient-, provider- and health facility-level factors associated with BP outcomes among patients receiving hypertension care at 15 health facilities in Kumasi, Ghana.

# Methods

## Setting

We conducted this study in the Kumasi metropolitan area, which is mostly urban and houses the capital city of the Ashanti region in Ghana [19]. According to the 2010 population and housing census in Ghana, the Kumasi metropolitan area had a population of 2,035,064 (36% of the Ashanti region's population) but increased to approximately 3,768,000 people in 2022 [20,21]. The urban population comprises 61.6% of the Ashanti region population and is growing at the second highest rate in Ghana (after the Greater Accra region). At the time of this study, the Kumasi metropolitan area was served by 263 health facilities, 145 of which were

clinics, three health centers, 53 hospitals, six training institutions, 55 maternity homes, and one regional hospital [22]. Of all the health facilities, 224 were privately owned, quasi-governmental organizations owned 18, 16 were government-owned, and five were owned by the Christian Health Association of Ghana (CHAG) [22].

## Design

We used a cross-sectional study design to explore the patient-, provider-, and health facility-level determinants of systolic and diastolic BP and the overall BP control. We followed the Strengthening the Reporting of Observational Studies in Epidemiology (STROBE) checklist [23] to ensure an elaborate and unbiased reporting and discussion of the findings (S1 Table).

**Power calculation and sample size.** The power analysis assumed an Intra Cluster Correlation (ICC) of 0.1 based on the systolic BP ICC reported by the Phone-based Intervention under Nurse Guidance after Stroke (PINGS) study interim results [24] and a BP control rate of 45.5% among hypertensive patients on treatment in the Ashanti region from a prior study [25]. With these assumptions and a sample of 254 patients, our mixed-effects logistic regression had 80% power to predict odds ratios of BP control of 1.92 or greater with continuous predictor variables with an alpha of 0.05. For dichotomous predictor variables, odds ratios of 3.31, 3.37, and 4.28 were to be detectable if the predictor variable had a 50/50, 40/60, and 20/80 split, respectively.

## Participants

We purposively selected 25 health facilities at different health system levels and with different ownerships. Fifteen of the health facilities agreed to participate in the study. Eligible health facilities: (1) provided hypertension service per the Ministry of Health structure; (2) qualified as government, quasi-governmental, private, or CHAG hospitals; and (3) had leadership willing to be part of the study. We recruited up to four clinical healthcare providers involved in hypertension management in each participating health facility's hypertension clinic or outpatient department (OPD). To be included, healthcare providers had to be: (1) nurses, physicians, physician assistants, or dietitians; (2) working in hypertension clinics or OPD; and (3) directly involved in managing patients with hypertension. While most nurses were dedicated to hypertension clinics or OPD, physicians, physician assistants, and dietitians also managed patients in other health facility departments.

Additionally, from each health facility, we recruited 15 patients with hypertension. To be included, patients had to (1) have been diagnosed and initiated on any kind of hypertension treatment; (2) be receiving ambulatory care at participating health facilities; and (3) be 18 years of age or older. Both male and female patients were eligible to participate in the study. Critically ill or hospitalized patients and pregnant women were not included in the study since their BP would be different from that of the other participants and not necessarily from the influence of the patient-, provider-, or health system factors under exploration by the current study.

We conducted the recruitment and data collection between April 15th and June 1st, 2022.

## Recruitment and data collection process

Of the 25 eligible health facilities we identified and invited to participate, 15 accepted to join the study and signed the memorandum of understanding (MOU). We traveled to the health facilities for data collection and dedicated a minimum of two days per health facility for data collection. After arriving at the health facility, a pair of trained enumerators, led by the ranking health facility leader, visited the different health facility units to collect facility-level data about

hypertension. We also invited all eligible healthcare providers at their health facilities to complete the online survey. Healthcare providers who were present on the day of data collection were likely to fill out the survey even though it was self-administered, with the possibility of filling it out at home. We encouraged providers to share the survey with their eligible colleagues who were not on duty during our visit to the health facility. Even though we invited clinical care providers from participating health facilities, providers were free to refuse to take the survey.

On the second day, the team recruited patients presenting for their usual hypertension care appointments. We worked with the health facilities that signed the MOU of this study to fix the data collection date on days when patients were usually scheduled for their hypertension follow-up appointments. In the morning, a research assistant explained to all patients (sitting in the waiting area) the purpose of the study and inclusion criteria. We then invited eligible patients to participate and enrolled those interested until we reached the desired number from each health facility. Those who accepted to join signed the consent forms before sitting for survey questions and blood pressure measurements that were part of the data collection. If we could not recruit the intended number of patients at a specific health facility, we scheduled an additional visit to the health facility for patient recruitment.

## Outcome of interest

The primary outcome was the ascertainment of patients' systolic and diastolic BP. We treated the systolic and diastolic BP as continuous variables. To measure patients' BP, we used the Omron 7 Series Upper Arm BP Monitor following the American Heart Association's guidelines for BP measurement [26]. After a minimum of five minutes of patient rest, we took three consecutive measures of BP with 1 minute between measurements. We used the average of the last two for BP measurement ascertainment.

*Secondary outcomes.* The secondary outcome was BP control as a binary variable. In accordance with Ghana hypertension treatment guidelines, a BP of <140/90 mmHg was considered controlled [27].

*Exposures of interest.* The exposures of interest were at the patient, healthcare provider, and health facility levels.

*Patient-level exposures*

We measured the patients' adherence to high BP treatment using the Hill-Bone Compliance to High Blood Pressure Therapy Scale (HB-HBP), a 14-item scale with an internal consistency reliability of 0.85 (S2 Table) [28]. HB-HBP has not been validated in Ghana but has been widely utilized in this setting [29]. HB-HBP has four choice options on the Likert scale ranging from "All of the Time" to "None of the Time," valid for 1 to 4 points respectively in that order, except for item 6, "How often do you make the next appointment before you leave the doctor's office?" (which required reverse coding). HB-HBP scores ranged from 14 to 56, with higher scores indicating behaviors of higher adherence to high BP therapy. HB-HBP consists of three sub-scales: (1) dietary sodium intake, three items with score ranging from 3 to12; (2) medication adherence, nine items with score ranging from 9 to 36; and (3) appointment keeping, two items with score ranging from 2 to 8. We treated the HB-HBP overall and subscale scores as continuous variables consistent with the scale-developer recommendations.

The other self-reported exposure variables that we measured at the patient level were the length of time with a hypertension diagnosis, which we treated as a binary variable (<five years, ≥five years), and travel duration to the health facility where they received hypertension care services as a categorical variable (< 30 minutes, 30 minutes to < one hour, and ≥ one hour) on any type of transportation (walking and public or private transportation).

*Healthcare provider-level exposures*

The provider-level predictors were the provider's knowledge of and attitude toward hypertension treatment guidelines. At the time of this study, patients in Ghana were not assigned a primary care provider who would ideally follow them up over time. They would rather meet the healthcare provider who was in the consulting office on the day of their appointment. For this reason, we could not link specific provider responses with a given patient's BP outcomes; hence, we could not ascertain the association between a patient's BP and the provider's knowledge score or attitudes toward hypertension guidelines.

We measured provider knowledge of hypertension treatment guidelines using a 23-item questionnaire developed by the investigation team (S3 Table). Five items were multiple choice questions with four response options, six were multiple choice questions with three response options plus "I don't know," and 14 items were true or false questions with "I don't know" as an additional response option. The respondent scored one point for each correct selection on the 23 items. We reported the total score, with higher scores indicating higher knowledge of hypertension guidelines. The psychometric properties of the tool have not been established.

To measure the providers' attitudes towards hypertension guidelines, we assessed their self-reported confidence in adhering to hypertension treatment guidelines. To measure their self-reported confidence, we asked them to rate on a scale of "Very confident" to "Not at all confident" their adherence to guidelines while (1) measuring BP, (2) assigning hypertension diagnosis, (3) educating patients with hypertension on healthy lifestyles, (4) prescribing antihypertensive medications, and (5) their overall management of patients with hypertension. In addition to this 5-point Likert scale, we had a "not applicable" option for tasks outside the provider's scope of practice.

*Health facility-level exposures*

We used the World Health Organization (WHO) Service Availability and Readiness Assessment Tool to collect health facility-level measurements of hypertension service readiness [30]. Health facility characteristics included bed capacity as a categorical variable (<50, 50 to <100 beds, and ≥100 beds), facility ownership as a categorical variable (government, private, or Christian Health Association of Ghana (CHAG) health facilities), and possession of a hypertension clinic (versus managing patients in the general OPD) as a binary variable. The bed capacity variable, though not directly related to outpatient management of patients with hypertension, was utilized as a proxy of the size of the health facility since all facilities included had in-patient service capacity. We later evaluated whether the size of the health facility was associated with hypertension outcomes.

The indicators related to the health workforce we measured were the patient-to-nurse ratio and patient-to-physician or physician-assistant ratio. To calculate the patient-to-clinician ratio, we divided the number of patients received in the OPD in one recent month by the number of registered and enrolled nurses or physicians (generalists and specialists) and physician assistants. We combined the physician and physician assistants because they were licensed to diagnose and prescribe antihypertensive medications in their scope of practice in Ghana. We categorized the patient-to-nurse ratio as < 20, 20 to < 40, and ≥ 40 patients per nurse per month, and the patient-to-physician or physician-assistant ratio <140, 140 to < 280, and ≥ 280 patients per physician (or assistant) per month.

The hypertension service readiness indicators we assessed were the availability of the 2017 Ghana Standard treatment guidelines [31] or the 2019 National Guidelines for the Management of Cardiovascular Diseases (first edition) [27]; patient educational materials consisting of brochures and risk charts; and BP measurement apparatus (functional automated BP device or a sphygmomanometer with a stethoscope). Additionally, we assessed the availability of all first-line antihypertensive medications (Lisinopril, Losartan, Amlodipine or Nifedipine, and

Hydrochlorothiazide or Bendroflumethiazide) and all basic laboratory exams essential for patients with hypertension, as required by Ghana's Standard Treatment Guidelines, 7[th] Edition [31], and the Essential Medicines List 7[th] Edition [32]. The routine basic laboratory exams for patients with hypertension include the following: (1) a valid dipstick for urine protein, glucose, and ketones (these are often measured by the same dipstick), (2) a glucometer for blood glucose, and valid test strips, and (3) hemoglobinometer or analyzer for hemoglobin testing.

*Covariates*

The patient-level covariates we explored included age, educational level, employment status, and marital status. We treated patient age as a continuous variable, biological sex as a binary variable (male, female), education as an ordinal variable (primary education or less, secondary education, and tertiary education or higher), employment status as a categorical variable (unemployed, employed, and retired), and marital status as a binary variable (married or cohabiting and single, separated or widowed).

The provider-level covariates we evaluated were age, sex, educational level, profession, and experience. We treated provider age as a continuous variable and sex as a binary variable (male, female), education as a categorical variable (associate degree or less, bachelor's degree, and post-graduate degree), and profession as a categorical variable (physicians (generalist, specialist, physician assistant), and nurses (enrolled and registered), and other clinical staff). We treated the experience as a categorical variable (<two years, two to <four years, and ≥four years).

## Analysis

We explored the patients' characteristics by their BP control status. We used means (SD) to report normally distributed continuous variables, median and interquartile ranges (IQR) for continuous data that are not normally distributed, and proportions and percentages for categorical variables. We used two-sample *t*-tests to compare the means of normally distributed continuous variables, Kruskal Wallis test to compare means and medians of assymetric data, and chi-square tests to compare the proportions of the categorical variables by the BP control status.

We used percentages to summarize provider characteristics and mean (SD) and median (IQR) to summarize provider scores of hypertension guidelines knowledge. We used Kruskal Wallis test to compare the scores of hypertension guidelines knowledge across the different categorical variables of provider characteristics. For the provider attitude towards hypertension guidelines, we reported the percentages of providers in each level of self-reported confidence in adherence to the guidelines. We used a stacked graph to report the providers' levels of self-confidence in adhering to hypertension treatment guidelines.

Since the health facility-level data were limited in the number of observations, we reported proportions (proportion of health facilities that met a certain standard of hypertension care) and percentages for categorical variables and the median and interquartile ranges (IQR) for continuous variables.

We used mixed-effects linear regression models to explore the patient- and health facility-level factors of systolic and diastolic BP and used mixed-effects logistic regression models to explore the patient- and health facility-level factors of BP control. We chose this model to account for possible clustering in BP outcomes across health facilities.

The selection of variables to include in fully adjusted models was guided by the WHO's health systems framework [33]. The first model (model 1) was unadjusted, and model 2 was fully adjusted for all key predictor variables and covariates separately for patient and facility-level factors. At the patient level, the fully adjusted model included the variables of compliance

with hypertension treatment (measured using HB-HBP), travel distance to the health facility, duration with hypertension diagnosis, and all sociodemographic characteristics as covariates. At the facility level, the fully adjusted model included the health facility characteristics (facility ownership, availability of hypertension clinic, and bed capacity) to account for service delivery and financing elements of the six building blocks of the health systems [33]; health workforce indicators (patient-to-clinician ratio); and hypertension service availability indicators (availability of all first-line antihypertensive medications). The distribution of data for these health facility-level variables—basic laboratory exams, BP devices, and hypertension guidelines, and patient educational materials—rendered the adjusted model unstable and were excluded from model 2. We reported the coefficients or odds ratio with the associated 95% confidence intervals. We considered associations with 0.05 or smaller $p$-values to be statistically significant. We used Stata/BE 17.0 for data analysis.

### Ethical consideration

The present study was approved by the committee on human research, publication, and ethics at Kwame Nkrumah University of Science and Technology–School of Medicine and Dentistry (Ref: CHRPE/AP/021/22) and the Johns Hopkins Medicine Institutional Review Board (IRB00218586). After securing the ethics committee approvals, we secured signed memoranda of understanding with each participating health facility detailing the study procedures, timelines, and potential individuals to be invited to the study. Prior to participating in the study, all patients and healthcare providers signed informed paper and electronic consent, respectively. We compensated patients and providers with 10 USD and 25 USD, respectively, for their time.

### Inclusivity in global research

Additional information regarding the ethical, cultural, and scientific considerations specific to inclusivity in global research is included in the (S1 Checklist)

## Results

### Patient characteristics

This study included 224 patients with hypertension with a mean (SD) age of 60.5 (12.7) years. Most participants, 182 (81%), were female. The mean (SD) systolic BP was 139 (20) mmHg, and diastolic BP was 85 (13) mmHg (Table 1). Almost half of the participants, 109 (49%), had controlled BP ($<140/90$ mmHg). A little over half of the patients (54%) were single, separated, or widowed, almost half of the patients (47%) were employed, and 199 (45%) of the patients had a tertiary or higher level of education. One hundred and forty-two (64%) patients had lived with a hypertension diagnosis for more than five years.

The overall mean (SD) score of compliance with high BP therapy was 50.39 (3.26) out of 56 possible maximum scores. The mean medication adherence score was 33.36 (2.21) out of 36; the mean appointment-keeping score was 7.12 (1.18) out of 8; and the mean diet-keeping score was 9.90 (1.66) out of 12.

### Provider characteristics

Healthcare providers (n = 67) were 32 (7) years of age on average (Table 2). Most healthcare providers, 53 (79%), were female, 40 (61%) had an associate degree or less, and the majority, 61 (91%), were enrolled or registered nurses. Of the healthcare providers, 32 (48%) had four or more years of experience, 39 (59%) worked in general OPD, and 44 (66%) reported that they had received training on hypertension management in the previous two years. The most

**Table 1. Patients' characteristics stratified by blood pressure control status (BP <140/90 mmHg), n (%) or mean (SD) (n = 224).**

| Patient level factors | Total n = 224 | Uncontrolled n = 115 | Controlled n = 109 | *p*-value |
|---|---|---|---|---|
| **Age (years), mean (SD)** | 60.50 (12.7) | 60.3 (11.3) | 60.8 (14.0) | 0.749 |
| **Sex** | | | | |
| *Male* | 42 (18.7%) | 21 (50%) | 21 (50%) | 0.847 |
| *Female* | 182 (81.3%) | 94 (51.7%) | 88 (48.3%) | |
| **Marital status** | | | | |
| *Married or cohabiting* | 104 (46%) | 50 (48%) | 54 (52%) | 0.363 |
| *Single/Separated/widowed* | 120 (54%) | 65 (54%) | 55 (46%) | |
| **Employment status** | | | | |
| *Unemployed* | 94 (42%) | 51 (54%) | 43 (46%) | 0.542 |
| *Employed* | 106 (47%) | 54 (51%) | 52 (49%) | |
| *Retired* | 24 (11%) | 10 (42%) | 14 (58%) | |
| **Education** | | | | |
| *Primary or less* | 65 (29%) | 35 (54%) | 30 (46%) | 0.869 |
| *Secondary level* | 59 (26%) | 29 (49%) | 30 (51%) | |
| *Tertiary or higher* | 199 (45%) | 51 (51%) | 49 (49%) | |
| **Duration of hypertension diagnoses** | | | | |
| *≤5 years* | 80 (36%) | 40 (50%) | 40 (50%) | 0.762 |
| *> 5 years* | 142 (64%) | 74 (51%) | 68 (49%) | |
| **Travel duration to the health facility** | | | | |
| *<30 minutes* | 90 (40%) | 41 (46%) | 49 (54%) | 0.125 |
| *30 to <1 hour* | 110 (49%) | 64 (58%) | 46 (42%) | |
| *≥1 hour* | 24 (11%) | 10 (42%) | 14 (58%) | |
| **Systolic Blood Pressure (mmHg), mean (SD)** | 139 (20) | 153 (17) | 123 (10) | <0.001 |
| **Diastolic Blood Pressure (mmHg), mean (SD)** | 85 (13) | 93 (11) | 76 (8) | <0.001 |
| Compliance with high BP therapy scores[1] | **Mean Score (SD)** | **Mean Score (SD)** | **Mean Score (SD)** | |
| **Diet (3 items), mean (SD)** | 9.90 (1.66) | 9.86 (1.63) | 9.94 (1.70) | 0.706 |
| **Medication (9 items), mean (SD)** | 33.36 (2.21) | 33.28 (2.39) | 33.45 (2.01) | 0.562 |
| **Appointment (2 items), mean (SD)** | 7.12 (1.18) | 7.17 (1.12) | 7.09 (1.25) | 0.643 |
| **Overall score (14 items), mean (SD)** | 50.39 (3.26) | 50.30 (3.01) | 50.49 (3.52) | 0.677 |

[1]Compliance with high blood pressure therapy scores was measured using the Hill-Bone Compliance with High Blood Pressure Therapy Scale [28]. Diet (three items assess dietary sodium consumption), Medication (9 items assess the likelihood of missing pills), and Appointment (two items assess how patients keep their medical appointments for hypertension care). Abbreviation: BP- Blood Pressure.

common type of training, however, was the job orientation and on-the-job training. In the context of hypertension care, job orientation is the type of training offered to employees newly hired or assigned to work at hypertension clinics or outpatient departments (which also manage patients with hypertension). These types of training varied in length, quality, and content from facility to facility. Further, the on-the-job training consisted of regular provider-initiated presentations on a specific topic and not necessarily structured training on hypertension management.

## Health facility characteristics

We conducted the study at 15 health facilities, among which almost half (7/15) had less than 50-bed capacity, and nearly half (7/15) were government-owned (Table 3). Eight (8/15) health facilities managed patients in the hypertension clinic; the remaining seven managed them in the general OPD.

**Table 2. Healthcare provider characteristics (n = 67).**

| Characteristic | Number (%) |
|---|---|
| **Mean Hypertension Knowledge Score (SD)** | 18.9 (1.9) |
| **Age in years, mean (SD)** | 32 (7) |
| **Sex** | |
| *Male* | 14 (21%) |
| *Female* | 53 (79%) |
| **Education (n = 66)** | |
| *Associate degree or less* | 40 (61%) |
| *Bachelor's degree* | 20 (30%) |
| *Post-graduate degree* | 6 (9%) |
| **Profession** | |
| *Physician/Doctors* | 6 (9%) |
| *Enrolled & Registered Nurses* | 55 (82%) |
| *Other clinical care providers* | 6 (9%) |
| **Experience** | |
| *< 2 years* | 19 (28%) |
| *2 to <4 years* | 16 (24%) |
| *≥4 years* | 32 (48%) |
| **Health facility department** | |
| *Hypertension Clinic* | 10 (15%) |
| *General OPD* | 39 (59%) |
| *Others* | 17 (26%) |
| **Received hypertension training in the last two years** | |
| *No* | 23 (34%) |
| *Yes* | 44 (66%) |
| **Type of hypertension training received (n = 44)** | |
| *Job orientation*[*1] | 13 (30%) |
| *On job training* | 15 (34%) |
| *Professional certification program* | 7 (16%) |
| *Conference or other hypertension training workshops* | 9 (20%) |

Abbreviations: OPD: Outpatient Department.

[*1] Job orientation: Training is offered to newly hired or appointed employees to work in general OPD or hypertension clinics. The training varies in content, quality, and length from facility to facility.

Regarding the health facilities' hypertension service readiness indicators, hypertension guidelines were available in 12/15 health facilities, patient educational materials were available in 9/15 health facilities, all basic laboratory tests relevant to hypertension were available in 11/15 health facilities, and all 15 health facilities had at least one functional BP measurement device. Furthermore, the median number of BP measurement devices was 2 (IQR: 2–4). Ten (10/15) health facilities had large cuffs for BP devices and none had a small cuff for BP devices.

All first-line antihypertensive medications were available at three (3/14) health facilities assessed for medication availability. One hundred (100) 30-day doses (IQR: 46–310) of Lisinopril 10 mg were available at 13/14 health facilities, and 21 (IQR: 0–209) had been dispensed the prior month (Table 4). The medications missing at most health facilities were Hydrochlorothiazide and Bendroflumethiazide—despite them being pivotal in the management of hypertension.

**Table 3. Characteristics of included health facilities in Ashanti Region, Ghana (n = 15).**

| Facility characteristics, n (%) or median (IQR) | Sites |
|---|---|
| **Facility size (by number of beds)** | |
| <50 beds | 7/15 (47%) |
| 50 to <100 beds | 5/15 (33%) |
| ≥100 beds | 3/15 (20%) |
| **Facility ownership** | |
| Government | 7/15 (47%) |
| Private | 5/15 (33%) |
| CHAG | 3/15 (20%) |
| **Patients treated in hypertension clinic** | |
| No (General OPD only) | 7/15 (47%) |
| Yes | 8/15 (53%) |
| **Personnel and training in cardiovascular disease management** | |
| Number of full-time nurses (registered and enrolled), median (IQR) | 35 (24–205) |
| Number of full-time doctors (generalists and specialists), median (IQR) | 9 (5–17) |
| Number of full-time paramedical clinicians, median (IQR) | 2 (1–7) |
| Patient/nurse ratio, median (IQR)–based on patient received in OPD last month | 26 (19–46) |
| Patient/physician ratio, median (IQR)–based on patient received in OPD last month | 235 (139–388) |
| **Hypertension Service Readiness Indicators, n (%)** | |
| Hypertension guidelines[*1] | 12/15 (80%) |
| Patient education materials | 9/15 (60%) |
| Basic laboratory exams available[*2] | 11/15 (73%) |
| All first-line antihypertensive medications available[*3] | 3/15 (20%) |
| At least one functional BP measurement device[*4] | 15/15 (100%) |
| Number of functional BP measurement devices, median (IQR) | 2 (2–4) |
| A medium-size cuff for BP devices was available | 15/15 (100%) |
| A large-size cuff for BP devices was available | 0/15 (0%) |
| A small cuff for BP devices was available | 10/15 (67%) |

Abbreviations: IQR–Interquartile Range; OPD–Outpatient Department; CHAG: Christian Health Association of Ghana; BP–Blood Pressure

[*1]Considered the 2017 national treatment standards or 2021 cardiovascular diseases guidelines.

[*2] Included five tests: Valid dipstick for urine protein, glucose, and ketones; glucometer for blood glucose and valid test strips; and hemoglobinometer or analyzer for hemoglobin testing.

[*3] All first-line medications considered according to Ghana Protocol: Lisinopril, Losartan, Amlodipine or Nifedipine, and Hydrochlorothiazide or Bendroflumethiazide.

[*4] An automatic blood pressure device or sphygmomanometer with a stethoscope was considered.

## Patient-, provider-, and facility-level predictors of systolic and diastolic BP

**Patient-level predictors of systolic and diastolic BP and BP control.** None of the explored patient-level factors was associated with systolic BP in the unadjusted and fully adjusted models (Tables 5 and 6). Longer travel duration to the health facility was associated with higher diastolic BP and lower odds of controlled BP in the fully adjusted but not in the unadjusted models (Tables 5 and 6). Higher scores of compliance with high BP therapy were associated with lower diastolic BP in both the unadjusted and fully adjusted models (Table 5). In the fully adjusted model, traveling for an hour or more to the health facility was associated with roughly 7 mmHg higher (Coefficient: 6.86; 95% CI: 0.92, 12.81) diastolic BP than

**Table 4. Availability of antihypertensive medications, n = 14.**

| Antihypertensive Class and Medications | Number of facilities with the medication in the last 30 days (n = 14) | 30-day doses in stock, Median (IQR), Range | 30-day doses dispensed in the previous 30 days, Median (IQR), Range |
|---|---|---|---|
| **Angiotensin-converting enzyme inhibitor** | | | |
| Lisinopril 10 mg | 13 | 100 (46–310), (0–3,606) | 21 (0–209), (0–372) |
| Lisinopril 5 mg | 6 | 122 (30–300), (13–1,092) | 58 (0–200), (0–1,400) |
| **Angiotensin receptor blocker** | | | |
| Losartan 100 mg | 6 | 230 (1–677), (0–5,760) | 177 (4–310), (0–1,620) |
| Losartan 50 mg | 12 | 198 (43–2,128), (0–5,760) | 118 (9–976), (0–1,920) |
| **Calcium channel blocker** | | | |
| Amlodipine 10 mg | 12 | 325 (159.5–1,699), (10–4,837) | 165 (60–552.5) (0–2,100) |
| Amlodipine 5 mg | 11 | 52 (24–120) (0–1,834) | 35 (7–70) (0–1,680) |
| Nifedipine 30 mg | 13 | 200 (60–1,260), (0–8,260) | 247 (20–840), (0–2,110) |
| Nifedipine 20 mg | 10 | 50 (30–95), (24–164) | 17 (0–90), (0–869) |
| **Diuretic** | | | |
| Spironolactone 50 mg | 4 | 65 (29–586), (0–1,092) | 35 (18.5–650), (7–1,260) |
| Spironolactone 25 mg | 2 | 49 (28–70), (28–70) | 10 (0–20), (0–20) |
| Hydrochlorothiazide 12.5 mg, count | 1 | 58 | 40 |
| Bendroflumethiazide 2.5 mg | 3 | 202 (151–284), (100, 366) | 80 (40–233), (0–366) |
| **Beta-blocker** | | | |
| Propranolol 40 mg, count | 1 | 112 | 28 |
| Atenolol 50 mg | 2 | 100 (42–159), (42–159) | 71 (20–122), (20–122) |
| Carvedilol 6.25 mg, count | 1 | 10 | 4 |
| **Central acting agent** | | | |
| Methyldopa 250 mg | 12 | 179.5 (100–500.5), (10–15,000) | 104.5 (38–183), (0–4,780) |
| **Vasodilator** | | | |
| Hydralazine 20 mg* | 4 | 35 (15–235), (10–420) | 11.5 (1.5–30), (0–40) |
| **Fixed Dose Combinations** | | | |
| Amlodipine / Valsartan / Hydrochlorothiazide 10.6 mg, count | 1 | 84 | 56 |
| Losartan/ Hydrochlorothiazide 50 mg, count | 1 | 617 | 14 |

Abbreviations: IQR–Interquartile Range.

traveling 30 minutes or less. Finally, a unit increase in overall HB-HBP scores was associated with a 0.5 mmHg lower (Coefficient: -0.55, 95% CI: -1.09, -0.01) diastolic BP. In the HB-HBP sub-scores, a unit increase in the diet (reduced sodium intake) sub-scores was associated with one unit lower (Coefficient: -1.11; 95% CI: -2.20, -0.02) diastolic BP in mmHg.

**Table 5. Mixed-effects linear regression of patient-level factors of systolic and diastolic blood pressure in Kumasi, Ghana (n = 221 patients, 15 health facilities).**

| | Model 1 | Model 2 |
|---|---|---|
| | ß (95% CI) | ß (95% CI) |
| **SYSTOLIC BLOOD PRESSURE** | | |
| Duration of hypertension diagnoses | | |
| ≤5 years | Ref. | Ref. |
| > 5 years | 2.58 (-3.06, 8.22) | 0.67 (-5.44, 6.78) |
| Travel duration to the health facility | | |
| <30 minutes | Ref. | Ref. |
| 30 to <1 hour | 2.15 (-3.59, 7.89) | 3.1 (-2.7, 8.91) |
| ≥1 hour | 3.96 (-5.34, 13.27) | 5.46 (-3.96, 14.87) |
| Compliance with high BP therapy[*1] | | |
| Diet (3 items) [*2] | 0.09 (-1.54, 1.73) | -0.47 (-2.22, 1.28) |
| Medication (9 items) [*2] | -0.78 (-2.00, 0.43) | -0.95 (-2.19, 0.29) |
| Appointment (2 items) [*2] | 1.38 (-0.93, 3.69) | 0.89 (-1.44, 3.23) |
| Overall score (14 items) | -0.17 (-0.99, 0.66) | -0.46 (-1.33, 0.41) |
| **DIASTOLIC BLOOD PRESSURE** | | |
| Duration of hypertension diagnoses | | |
| ≤5 years | Ref. | Ref. |
| > 5 years | -2.69 (-6.33, 0.94) | -0.98 (-4.75, 2.79) |
| Travel duration to the health facility | | |
| <30 minutes | Ref. | Ref. |
| 30 to <1 hour | 2.72 (-1.01, 6.45) | **3.75 (0.12, 7.38)** [†] |
| ≥1 hour | 4.27 (-1.80, 10.34) | **6.86 (0.92, 12.81)** [†] |
| Compliance with high BP therapy scores[*1] | | |
| Diet (3 items) [*2] | **-1.23 (-2.29, -0.17)** [†] | **-1.11 (-2.20, -0.02)** [†] |
| Medication (9 items) [*2] | **-0.97 (-1.75, -0.19)** [†] | -0.72 (-1.49, 0.06) |
| Appointment keeping (2 items) [*2] | 0.41 (-1.10, 1.93) | 0.51 (-0.97, 2.00) |
| Overall score (14 items) | **-0.70 (-1.23, -0.17)** [††] | **-0.55 (-1.09, -0.01)** [†] |

Abbreviations: BP: Blood Pressure; CI: Confidence Interval.

[*1] Compliance with high Blood Pressure therapy score was measured using the Hill-bone compliance with high blood pressure therapy scale [28]. Diet (three items assess dietary sodium consumption), Medication (9 items assess the likelihood of missing pills), and Appointment keeping (two items assess how patients keep their medical appointments for hypertension care).

Model 1: Unadjusted; Model 2: Adjusted for all other variables (except Hillbone treatment adherence sub-scores) plus age, sex, marital status, employment, education, and duration of hypertension diagnoses.

[*2] In Model 2, we adjusted for all other variables (except other Hillbone treatment adherence sub-scores and total score) plus age, sex, marital status, employment, education, and duration of hypertension diagnoses.

[†]$p < 0.05$

[††]$p < 0.01$.

Though it was not significant in the unadjusted model, the duration of travel to the health facility was significant in the fully adjusted mixed-effects logistic regression model. Patients who traveled 30 minutes to an hour to the health facility were half as likely to have controlled BP than their counterparts who traveled less than 30 minutes (OR: 0.51; 95% CI: 0.28, 0.92).

**Provider-level predictors of BP outcomes.** The mean provider knowledge score on hypertension guidelines was 18.9 (1.9) on a 23-point scale, and the median (IQR) score was 19 (15–22). In association analyses, no provider characteristics were associated with the provider

**Table 6. Mixed-effects logistic regression of patient-level factors of controlled blood pressure in Kumasi, Ghana (n = 221 patients, 15 health facilities).**

|  | Model 1 | Model 2 |
|---|---|---|
|  | Odds Ratio (95% CI) | Odds Ratio (95% CI) |
| Duration of hypertension diagnoses |  |  |
| ≤5 years | Ref. | Ref. |
| > 5 years | 0.91 (0.52, 1.59) | 0.84 (0.45, 1.56) |
| Travel duration to the health facility |  |  |
| <30 minutes | Ref. | Ref. |
| 30 to <1 hour | 0.60 (0.34, 1.05) | **0.51 (0.28, 0.92)** [†] |
| ≥1 hour | 1.17 (0.47, 2.91) | 1.03 (0.4, 2.67) |
| Compliance with high BP therapy scores*[1] |  |  |
| Diet (3 items) *[2] | 1.03 (0.88, 1.20) | 0.99 (0.82, 1.18) |
| Medication (9 items) *[2] | 1.04 (0.92, 1.17) | 1.03 (0.91, 1.17) |
| Appointment (2 items) *[2] | 0.95 (0.75, 1.19) | 0.98 (0.78, 1.24) |
| Overall score (14 items) | 1.02 (0.94, 1.10) | 1.01 (0.92, 1.1) |

Abbreviations: BP- Blood Pressure; CI- Confidence Interval.

*[1] Compliance with high Blood Pressure therapy score was measured using the Hill-bone compliance with high blood pressure therapy scale [28]. Diet (three items assess dietary sodium consumption), Medication (9 items assess the likelihood of missing pills), and Appointment keeping (two items assess how patients keep their medical appointments for hypertension care).

Model 1: Unadjusted; Model 2: Adjusted for all other variables (except Hillbone treatment adherence sub-scores) plus age, sex, marital status, employment, education, and duration of hypertension diagnoses.

*[2] In Model 2, we adjusted for all other variables (except other Hillbone treatment adherence sub-scores and total score) plus age, sex, marital status, employment, education, and duration of hypertension diagnoses.

[†] $p<0.05$

[††] $p<0.01$.

knowledge scores. Three-quarters (75% of the 64) of respondents reported being very or moderately familiar with hypertension treatment guidelines (Fig 1).

Regarding the nurses and other clinical staffs' (other than medical doctors) attitudes towards hypertension guidelines, almost 100% of the respondents were "very confident" in their adherence to the guidelines while measuring BP; roughly 80% were "very confident" in providing counseling on healthy lifestyles (Fig 2). Only half of the respondents were "very

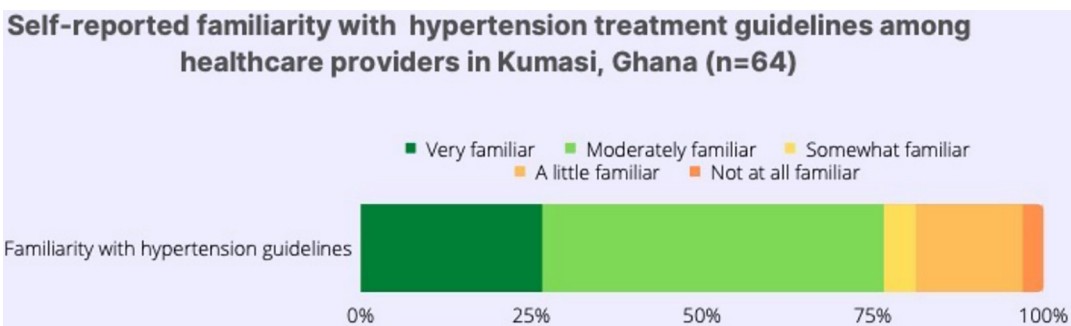

**Fig 1. Self-reported familiarity with hypertension treatment guidelines among healthcare providers in Kumasi, Ghana (n = 64).**

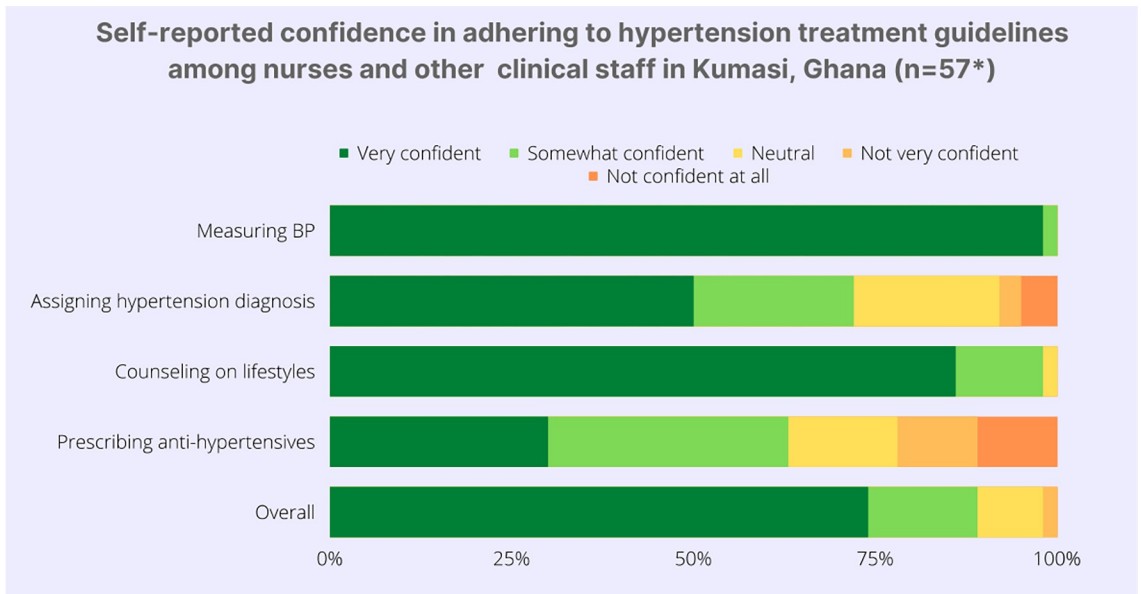

**\*Number of respondents varies by those who reported ever involved in performing the specific activity (Measuring BP: 55; Assigning diagnosis: 40; Counselling on lifestyles: 44; Prescribing antihypertensive med.: 27; Overall management of hypertension: 53)**

**Fig 2. Self-reported confidence in adhering to hypertension treatment guidelines among nurses and other clinical staff (except medical doctors) in Kumasi, Ghana (n = 57).**

confident" in diagnosing hypertension, and even fewer, roughly 30%, were confident in prescribing antihypertensive medications. Overall, 75% of the nurses and other clinical staff (except medical doctors) were very confident in adhering to hypertension guidelines while managing patients with hypertension.

*Health facility level predictors of systolic and diastolic BP and BP control.* At the health facility level, private facility ownership was associated with high systolic and diastolic BP in the fully adjusted but not the unadjusted models (Table 7). A higher patient-to-physician or physician-assistant ratio was associated with higher systolic BP and lower odds of controlled BP in both the unadjusted and fully adjusted models and with higher diastolic BP in the fully adjusted model only (Tables 7 and 8).

Compared to private health facilities, receiving care at government health facilities was associated with a 14 mmHg (Coefficient: -13.89; 95% CI: -23.99, -3.79) and 10 mmHg (Coefficient: -9.80; 95% CI: -16.08, -3.50) lower systolic and diastolic BP, respectively.

Receiving care at health facilities where a physician or physician assistant consulted 140 to 280 patients per month in OPD was associated with a 21 mmHg (Coefficient: 21.34; 95% CI: 8.94, 33.74) and 14 mmHg (Coefficient: 14.27; 95% CI: 6.54, 21.99) higher systolic and diastolic BP, respectively, than where a physician or physician assistant consulted less than 140 patients per month in OPD. At facilities where a physician or physician assistant consulted 280 or more patients per month, there was a 25 mmHg (Coefficient: 25.21; 95% CI: 8.51, 41.92) average higher systolic BP.

Patients receiving care at health facilities where a physician or physician assistant consulted 140 to 280 patients per month were five times less likely (OR: 0.19; 95% CI: 0.05, 0.72) to have controlled BP than those who received care where a physician or physician assistant consulted less than 140 patients per month.

**Table 7. Mixed-effects linear regression of health facility factors of systolic and diastolic blood pressure in Kumasi, Ghana (n = 221 patients, 15 health facilities).**

| | Model 1 | Model 2 |
|---|---|---|
| | ß (95% CI) | ß (95% CI) |
| **SYSTOLIC BLOOD PRESSURE** | | |
| Facility ownership | | |
| *Private* | Ref. | Ref. |
| *Government* | 2.82 (-3.76, 9.39) | **-13.89 (-23.99, -3.79)** [††] |
| *CHAG* | 4.36 (-3.17, 11.90) | **-17.10 (-30.58, -3.62)** [†] |
| Hypertension clinic | | |
| *General OPD only* | Ref. | Ref. |
| *Yes* | **-5.88 (-11.23, -0.52)** [†] | -2.88 (-9.04, 3.27) |
| Facility size by bed count | | |
| *<50 beds* | Ref. | Ref. |
| *50 to <100 beds* | 3.41 (-2.97, 9.80) | -1.33 (-9.85, 7.19) |
| *≥100 beds* | -0.62 (-8.01, 6.77) | -7.70 (-17.82, 2.41) |
| All first-line antihypertensive medications available | | |
| *No* | Ref. | Ref. |
| *Yes* | 2.02 (-5.24, 9.28) | -5.31 (-17.85, 7.22) |
| Patient-to-nurse ratio (patients per month per nurse) | | |
| *<20* | Ref. | Ref. |
| *20 to <40* | 4.14 (-2.65, 10.94) | -5.95 (-15.22, 3.31) |
| *≥40* | 3.83 (-3.86, 11.53) | 0.68 (-8.68, 10.05) |
| Patient-to-physician or physician-assistant ratio (patients per month per clinician) | | |
| *<140* | Ref. | Ref. |
| *140 to <280* | **6.34 (0.09, 12.59)** [†] | **21.34 (8.94, 33.74)** [††] |
| *≥280* | 6.81 (-0.11, 13.72) | **25.21 (8.51, 41.92)** [††] |
| **DIASTOLIC BLOOD PRESSURE** | | |
| Facility ownership | | |
| *Private* | Ref. | Ref. |
| *Government* | 2.96 (-1.97, 7.88) | **-9.80 (-16.08, -3.50)** [††] |
| *CHAG* | 1.01 (-4.67, 6.69) | -5.39 (-13.79, 3.00) |
| Hypertension clinic | | |
| *General OPD only* | Ref. | Ref. |
| *Yes* | -3.91 (-7.93, 0.11) | -2.12 (-5.96, 1.71) |
| Facility size by bed count | | |
| *<50 beds* | Ref. | Ref. |
| *50 to <100 beds* | -0.70 (-5.75, 4.36) | 1.35 (-3.96, 6.66) |
| *≥100 beds* | -0.84 (-6.72, 5.05) | -2.15 (-8.46, 4.15) |
| All first-line antihypertensive medications available | | |
| *No* | Ref. | Ref. |
| *Yes* | -0.10 (-5.60, 5.40) | -7.04 (-14.86, 0.762) |
| Patient-to-nurse ratio (patients per month per nurse) | | |
| *<20* | Ref. | Ref. |

*(Continued)*

**Table 7.** (Continued）

| | Model 1 | Model 2 |
|---|---|---|
| | ß (95% CI) | ß (95% CI) |
| *20 to <40* | 0.87 (-4.45, 6.19) | **-5.93 (-11.70, -0.15)** † |
| *≥40* | 0.25 (-5.77, 6.27) | -2.41 (-8.25, 3.42) |
| Patient-to-physician or physician-assistant ratio (patients per month per clinician) | | |
| *<140* | Ref. | Ref. |
| *140 to <280* | 4.46 (-0.11, 9.03) | **14.27 (6.54, 21.99)** †† |
| *≥280* | 0.05 (-5.01, 5.10) | 8.73 (-1.68, 19.14) |

Abbreviations: CHAG: Christian Health Association of Ghana; OPD: Outpatient Department; BP: Blood Pressure.

Model 1: Unadjusted; Model 2: *Adjusted for all other variables plus age, sex, education, and employment status.

†p<0.05

††p<0.01.

**Table 8. Mixed-effects logistic regression of health facility factors of blood pressure control in Kumasi, Ghana (n = 221 patients, 15 health facilities).**

| | Model 1 | Model 2 |
|---|---|---|
| | Odds Ratio (95% CI) | Odds Ratio (95% CI) |
| Facility ownership | | |
| *Private* | Ref. | Ref. |
| *Government* | 1.15 (0.63, 2.11) | 1.09 (0.36, 3.33) |
| *CHAG* | 0.81 (0.41, 1.63) | 0.98 (0.23, 4.18) |
| Hypertension clinic | | |
| *General OPD only* | Ref. | Ref. |
| *Yes* | 1.451 (0.86, 2.46) | 1.18 (0.62, 2.26) |
| Facility size by bed count | | |
| *<50 beds* | Ref. | Ref. |
| *50 to <100 beds* | 0.81 (0.44, 1.47) | 0.77 (0.31, 1.89) |
| *≥100 beds* | 1.09 (0.54, 2.16) | 2.470 (0.83, 7.33) |
| All first-line antihypertensive medications available | | |
| *No* | Ref. | Ref. |
| *Yes* | -0.09 (-0.74, 0.55) | 3.71 (0.98, 14.10) |
| Patient-to-nurse ratio (patients per month per nurse) | | |
| *<20* | Ref. | Ref. |
| *20 to <40* | 0.747 (0.40, 1.39) | 1.49 (0.55, 3.99) |
| *≥40* | 0.75 (0.370, 1.51) | 0.70 (0.26, 1.87) |
| Patient-to-physician or physician-assistant ratio (patients per month per clinician) | | |
| *<140* | Ref. | Ref. |
| *140 to <280* | **0.50 (0.27, 0.94)** † | **0.19 (0.05, 0.72)** † |
| *≥280* | 0.67 (0.34, 1.34) | 0.38 (0.06, 2.31) |

Abbreviations: CHAG-Christian Health Association of Ghana; OPD-Outpatient Department; BP-Blood Pressure.

Model 1: Unadjusted; Model 2: *Adjusted for all other variables plus age, sex, education, and employment status.

†p<0.05

††p<0.01.

## Discussion

The purpose of this study was to explore the health facilities' readiness for hypertension management and assess the patient-, provider-, and health facility-level factors associated with systolic and diastolic BP and BP control in Ghana. Most health facilities had hypertension treatment guidelines, at least one functional BP measurement device with a median of two functional devices, and all basic laboratory investigations. Conversely, the availability of patient educational materials and all first-line antihypertensive medications, as recommended by Ghana policy, was sub-optimal. We found that nearly half of the patients with hypertension had a controlled BP. Patients with shorter travel duration to the health facility and higher scores of treatment adherence were more likely to have better diastolic BP outcomes. Providers had high knowledge scores of hypertension guidelines and reported being very confident in adhering to them in practice. At the health facility level, receiving care at government health facilities and a low patient-to-physician or physician-assistant ratio were associated with better BP outcomes. Our study found a high rate of uncontrolled BP given that the participants were already on treatment, and we identified potential factors of focus at the patient and health facility levels for improving BP outcomes.

The findings of a high prevalence of uncontrolled hypertension among patients on treatment can be corroborated in the literature. Previous studies reported BP control among patients on treatment ranging from 23% up to 49% [34,35]. The 2019 May Measurement Month study reported a BP control prevalence of 49% among patients on treatment [30]. The latter study was conducted in Kumasi, recruited from the community, had roughly younger (mean age of 51 years) and a balanced mix of male and female participants compared to the current study [34]. Conversely, a study that recruited patients from two hospitals in greater Accra reported a 23% BP control level [35]. Studies recruiting patients in healthcare settings may report lower rates of BP control because people tend to seek care when they feel sick—possibly from complications of uncontrolled BP. For additional context, only 30% of patients aware of their hypertension status were initiated on treatment [3]. Since only 35% of people with hypertension are diagnosed, the overall estimate of patients with controlled BP narrows down to 6%, further demonstrating a wide gap in hypertension management [3].

We found that traveling for an hour or more to a health facility for hypertension care was associated with roughly 7 mmHg higher diastolic BP. A significant challenge with hypertension care in Ghana is that hypertension care services are centralized at district and referral hospitals [36]. Since there are no guidelines for managing patients with hypertension at the low levels of the health systems (community-level health facilities), and systems such as mobile clinics or digital healthcare are absent, patients must travel long distances to hospitals for their regular hypertension follow-up appointments [36]. In Rwanda, a similarly low-income SSA country, a study that decentralized care from district hospitals to community health centers demonstrated that patients achieved as high a level of controlled BP at community health centers as they did when they received care at hospitals, yet saving patients time and money which makes accessible hypertension services a sustainable strategy of care [37]. Strategies for improving geographical access to hypertension services are very much needed in Ghana.

In the present study, we found that higher hypertension treatment adherence scores were associated with lower diastolic BP but not with controlled BP. These findings corroborate the previous literature. In their study, Sarfo and colleagues (2020) showed that each one-unit increase in the scores of treatment adherence was associated with a quarter-unit decline in diastolic BP [38]. Multiple factors at the patient, provider, and health facility levels influence the patient's adherence to treatment. With the understanding that hypertension is a chronic condition requiring ongoing treatment, at the patient level, some of the reasons that hinder

compliance with treatment include forgetfulness, alcohol consumption and smoking behaviors, and a lack of motivation [39,40]. The provider factors include competence in educating patients on their health situation, treatment goals, and overall knowledge of hypertension guidelines. At the health facility and health systems level, medications' unavailability, inaccessibility, and unaffordability are major barriers to patients' adherence to treatment [39]. The success of any strategy to manage hypertension will, in many ways, require finding measures for addressing the factors of non-compliance with hypertension treatment.

Healthcare providers had ample knowledge scores on hypertension guidelines and reported a high overall self-confidence in adhering to guidelines on tasks that were in their scope of practice. These findings contrast the literature regarding the healthcare providers' perceptions regarding barriers to hypertension management in Ghana, which allude mostly to the lack of training opportunities [41,42]. The present findings could, however, be skewed by the social desirability bias. A qualitative interview with healthcare providers at community health centers in the Upper East Region of Ghana reported that providers felt that they had inadequate knowledge on screening, treatment, and prescribing medications for cardiovascular diseases, including hypertension, and expressed the need for regular training sessions, workshops, and mentorship opportunities on cardiovascular diseases management [36]. In our findings, three out of five healthcare providers reported that they had received training in hypertension management in the last two years, and the most common type of training available for healthcare providers was the job orientation for new hires and on-the-job presentations rather than structured training sessions for hypertension management. All clinicians involved in the management of patients with hypertension should receive regular and structured hypertension and overall cardiovascular disease management training and mentorship.

We found that at facilities where a physician or physician assistant consulted 280 or more patients per month, patients had an average of 25-mmHg higher systolic BP and were five times less likely to have controlled BP. Patient overload results in rushed consultations and inadequate time for appropriate patient needs assessment for treatment adjustment. Like many other LMICs, Ghana's health policy still does not permit nurses to diagnose and prescribe antihypertensive medications independently [36,43]. Multiple studies have successfully trained nurses to diagnose hypertension and prescribe medications, and patients have expressed positive views on this approach [36,42,44,45]. In our study, a higher number of nurses was not associated with better BP outcomes, and this could pertain to the fact that those nurses were not trained to support the diagnosis and medication prescription for hypertension. Policymakers in Ghana should consider shifting hypertension care tasks to nurses as overwhelmingly supported by the available evidence to address the shortage of clinicians, especially physicians, in hypertension management.

We found that receiving care at private health facilities was associated with poor systolic and diastolic BP outcomes. These findings are counterintuitive, given that most private health facilities were created because of inefficiencies in public health facilities and were often reported to have shorter wait times, better patient-provider relationships, and higher overall user satisfaction [46]. While the true reason for poor performance in hypertension management at private health facilities in Ghana is not known, we think that the key reasons included the lack of prioritization and investment in team-based care and the high cost of care. Team-based care involves having multiple healthcare providers across health facilities collaborate to manage patients, and previous studies have demonstrated that it is effective in hypertension management [47,48]. Yet, to minimize expenses, private health facilities de-emphasize the need to hire all key healthcare providers such as pharmacists, dieticians, and community health workers, all of whom are key personnel needed for team-based hypertension care. Further, the cost of care is usually higher in private than in public health facilities and is associated

with poor hypertension outcomes [49,50]. Future studies should explore the exact reasons for low performance in hypertension management by private health facilities in Ghana and devise appropriate solutions.

Our study comes with some limitations. First, using a cross-sectional design, we could not longitudinally assess the change in BP or the achievement of BP control and associated patient-, provider-, and facility-level factors. Second, our study was conducted at a few health facilities, which did not allow sufficient statistical power to detect small associations between facility-level predictors and the outcomes of interest. Third, the sampling of health facilities, providers, and patients did not use the ideal random sampling strategy. Despite our attempt to purposively choose a mix of different types of health facilities, the lack of random sampling could have led to us selecting health facilities, providers, and patients who were different from the ones who were not selected to participate or did not choose to participate. These factors could affect the external validity of our findings. Fourth, in Ghana and many other countries in SSA, there are no systems for patients to maintain one primary healthcare provider; hence, we could not link patients' outcomes with individual healthcare providers. We, therefore, aggregated variables of provider-level characteristics, knowledge scores, and self-reported adherence to hypertension guidelines but could not link them directly to patients' BP outcomes.

In additional limitations, some assessments of the patient- and provider-level predictors were based on self-reported measures, which could be associated with recall and social desirability bias. Last, the Service Availability and Readiness Assessment tool we used to measure health facility data has not been extensively utilized in other settings to measure hypertension service readiness, which limits the comparison of findings across studies and settings. Yet, the Service Availability and Readiness Assessment tool that we utilized has been expert-validated in Nigeria [51].

Despite the listed limitations, our study has major strengths in that it incorporates a broader viewpoint (from the patient up to the facility levels of care) on the factors associated with hypertension service provision and patients' BP outcomes. Our findings confirm prior findings about patient-level factors of BP outcomes and highlight the problems at the health system level that are essential for consideration by policymakers and public health professionals.

## Conclusion

In conclusion, hypertension control among patients with hypertension on treatment remains low. Patients with a long duration of travel to the health facility, receiving care at private health facilities, and where there are fewer physicians or physician assistants were more likely to have poor BP outcomes. Healthcare providers expressed high confidence in adhering to guidelines. In Ghana, interventions targeting the patient-, provider-, and health system levels will be needed to address the hypertension burden successfully. Specifically, in addition to addressing patient-level factors that promote adherence to behavioral and medical treatment, the public health experts and policymakers in Ghana should find ways to decentralize hypertension care services, institute interventions that address the shortage of physician and physician-assistants, and identify and address the root cause of underperformance among private health facilities. A larger-scale study should explore hypertension services' availability and readiness in urban and rural health facilities.

## Supporting information

**S1 Checklist. Inclusivity in global research.**
(DOCX)

**S1 Table. STROBE checklist.**
(DOCX)

**S2 Table. Provider knowledge and adherence to hypertension guidelines.**
(PDF)

**S3 Table. Hill-Bone Compliance to High Blood Pressure Therapy Scale (HB-HBP).**
(DOCX)

## Acknowledgments

We thank all the health facilities in Kumasi, Ghana, healthcare professionals, and patients who participated in the ADHINCRA study.

## Author Contributions

**Conceptualization:** Samuel Byiringiro, Yvonne Commodore-Mensah, Shadrack Assibey, Cheryl R. Himmelfarb.

**Data curation:** Samuel Byiringiro, Thomas Hinneh, Shadrack Assibey.

**Formal analysis:** Samuel Byiringiro, Nancy Perrin.

**Funding acquisition:** Samuel Byiringiro, Yvonne Commodore-Mensah, Fred Stephen Sarfo, Cheryl R. Himmelfarb.

**Investigation:** Samuel Byiringiro.

**Methodology:** Samuel Byiringiro, Yvonne Commodore-Mensah, Fred Stephen Sarfo, Nancy Perrin, Cheryl R. Himmelfarb.

**Project administration:** Samuel Byiringiro, Thomas Hinneh, Shadrack Assibey.

**Resources:** Samuel Byiringiro, Fred Stephen Sarfo.

**Software:** Samuel Byiringiro.

**Supervision:** Yvonne Commodore-Mensah, Fred Stephen Sarfo, Nancy Perrin, Cheryl R. Himmelfarb.

**Validation:** Thomas Hinneh, Yvonne Commodore-Mensah, Jill Masteller, Fred Stephen Sarfo, Nancy Perrin, Cheryl R. Himmelfarb.

**Visualization:** Samuel Byiringiro.

**Writing – original draft:** Samuel Byiringiro.

**Writing – review & editing:** Samuel Byiringiro, Thomas Hinneh, Yvonne Commodore-Mensah, Jill Masteller, Fred Stephen Sarfo, Nancy Perrin, Shadrack Assibey, Cheryl R. Himmelfarb.

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
