## [Decision Letter · Decision Letter 0]

10 Sep 2023

PGPH-D-23-01066

Exploring patient-, provider-, and health facility-level determinants of blood pressure among patients with hypertension: A multicenter study in Ghana

Dear Dr. Byiringiro,

Thank you for submitting your manuscript to PLOS Global Public Health. After careful consideration, we feel that it has merit but does not fully meet PLOS Global Public Health’s publication criteria as it currently stands. Therefore, we invite you to submit a revised version of the manuscript that addresses the points raised during the review process.

We look forward to receiving your revised manuscript.

Kind regards,

Nasheeta Peer

Academic Editor

Journal Requirements:

Additional Editor Comments (if provided):

Abstract

Line 32: Please add ‘years’ after “…mean (SD) 32 age of 32 (7)…” and …”age of 60.5 (12.7).“

Line 38: Please replace ‘less’ with ‘lower systolic BP’.

Background

Line 48: Please remove ‘chronic’

Line 54: Please improve the syntax: “…ill-equipped with frail health systems that are not prepared…” – remove “ill”; rather replace ‘not prepared’ with ‘unable to’

Line 75: I would suggest that healthcare system (adequate equipment, medication availability, etc.) and healthcare provider (provider inertia, inadequate training, etc.) factors should be considered separately, please.

Line 80: “…the facilities for physical exercises, and affordable foods for healthy diets…” I would suggest that these environmental factors are largely influenced by government policy rather than it being the responsibility of the community. Please rephrase.

Similarly, the patient support groups (line 78), while based in the community, would be an extension of the health care system, or has it been established by an independent NGO?

Line 81-82: This is vague: “…worsening status of hypertensive patients.” Please be specific – is it inadequate hypertension control or the development of comorbidities, etc?

Please remove the repetition: Lines 56-62 and Lines 81-97.

Line 98: Please add ‘associated with’ as follows: “…facility-level factors associated with systolic…”

Please tighten, shorten and focus the Background section by clearly articulating the key areas influencing hypertension care (policy and healthcare systems, health care provider and patient factors) and briefly touching on the related variables (medicine shortages, training of staff, etc.). It is currently quite long, somewhat repetitive and disjointed.

Methods

Please clarify the reason for the convenience sampling design vs. a random sampling technique of the health facilities.

How were healthcare providers and patients selected?

What were the participant (patients and healthcare providers) inclusion and exclusion criteria?

Did all healthcare providers who were interviewed treat patients with hypertension? Did they treat only patients with hypertension at a dedicated hypertension clinic or did they treat all outpatients?

Line 196: Please explain how “bed capacity” contributes directly to hypertension care and BP control? Also, most patients are generally treated at a clinic level.

Line 219: How is “hemoglobinometer or analyzer for hemoglobin testing” related to optimal hypertension care? Please clarify or remove.

Line 223: It should be Thiazide Diuretics.

Analysis: What statistical package was used, please?

Line 260: How were the “hypertension adherence” variables assessed, please?

Line 281-282: Was the denominator all participants with hypertension?

Line 286-287: ‘Status’ is repeated. Please remove ‘n (%) or mean (SD) (N = 224) n’ from the title and incorporate into the table – Table 1 and other tables.

Table 1: Please explain diet, medication and appointment in the footnote.

Table 2: What does ‘job orientation’ relate to in the context of hypertension training, please?

Please provide more details on ‘On job training’.

Line 290-291: Please clarify if each score was considered adequate, poor or good.

Lines 304-305 and Tables 3 and 4: Again, please clarify the importance of ‘bed capacity’ in hypertension management and control.

Table 3: Were “Essential antihypertensive medications” all first line therapy, please?

Line 317: Did all “training on cardiovascular diseases” include hypertension management training?

Line 323: Please provide examples of “second-line antihypertensive medication”.

Lines 322-328: Please discuss 1st line then 2nd line medication; not interchangeably.

Line 332-333: Please include the direction of the associations.

Line 338: replace ½ with 0.5 mmHg.

Lines 339-340: Please describe the “diet control scores” – was it related to salt intake specifically/only, etc.?

Line 351: Is the knowledge score considered good or poor?

Line 353: “Two-thirds (75%...” – please correct – 75% is three-quarters. Do the authors consider this proportion adequate?

Discussion

This section is overly long and unfocused.

After the summary of your findings, please discuss whether this study has adequately identified the variables contributing to hypertension control (good or poor control in this study?) in this population.

What are the public health implications of these study findings?

What can or should be done to address the challenges and improve hypertension control?

Line 400: Please rephrase: “…factors associated with systolic and…”

Line 402: Please replace ‘exams’ with ‘investigations’.

Line 405: Do you mean education level?

For example, (lines 490-491): is it adequate for a single health care provider to receive hypertension management training seeing that it did not improve hypertension control? As mentioned, the discussion is unfocused; it needs to be targeted to the findings of this study with tangible and practicable recommendations made for improving hypertension care (Lines 478-500). This applies to the entire Discussion section, please.

Lines 522-29: How would these structural changes directly impact hypertension care. Please link your discussion directly/tangibly to hypertension care.

Line 530-542: Please link cost of care to this study findings. Otherwise, mention as a limitation if it has not been addressed and remove this text.

Lines 554-556: Has the Service Availability and Readiness Assessment tool been validated to use for hypertension/BP control?

Lines 559-561: Please explain/substantiate how “This study provides data that is vital to entities looking to explore the multi-level factors…”

Line 564: Please explain or rephrase “facility type of management”.

Lines 568-572: I believe that this topic has already been extensively explored in the literature. Please remove/rephrase. Where are the specific research gaps?

Reviewers' comments:

Reviewer's Responses to Questions

**Comments to the Author**

1. Does this manuscript meet PLOS Global Public Health’s publication criteria? Is the manuscript technically sound, and do the data support the conclusions? The manuscript must describe methodologically and ethically rigorous research with conclusions that are appropriately drawn based on the data presented.

Reviewer #1: Yes

Reviewer #2: Yes

Reviewer #3: No

Reviewer #4: Yes

2. Has the statistical analysis been performed appropriately and rigorously?

Reviewer #1: Yes

Reviewer #2: Yes

Reviewer #3: I don't know

Reviewer #4: Yes

3. Have the authors made all data underlying the findings in their manuscript fully available (please refer to the Data Availability Statement at the start of the manuscript PDF file)?

Reviewer #1: Yes

Reviewer #2: Yes

Reviewer #3: No

Reviewer #4: Yes

4. Is the manuscript presented in an intelligible fashion and written in standard English?

Reviewer #1: Yes

Reviewer #2: Yes

Reviewer #3: Yes

Reviewer #4: No

5. Review Comments to the Author

Reviewer #1: Congratulations to the authors for such a well planned and executed study of relevance for the management of hypertension

It reads very well and captures the salient findings of what the authors set out to find.

I have no comments than to congratulate the authors on the study.

Reviewer #2: This was a very thorough review of issues related to hypertension in Ghana. Some topics that you may also want to expand on is there any additional risk factors for hypertension in Ghana, i.e. hyperlipidemia, diabetes, diet, exercise? Also are there any rates on the amount of cardiovascular disease/stroke related to untreated hypertension?

If it is possible to expand on what changes could be suggested at the end of the paper, I think that may also be helpful.

What can Ghana do to improve rates of hypertension: is it to increase public health clinics? Given the private clinics do not seem to be having much success, should they be monitored more? Should there be a decrease number of patients treated per healthcare professional at public clinics? Should nurses be allowed to do prescribing and have community health workers follow-up on blood pressure checks in patients homes? Should there be teaching provided in workplaces and schools regarding dietary management and exercise and what blood pressure numbers correspond to hypertension?

In the conclusion, you may want to have some specific steps for how to improve the current rates of diagnosis and treatment of hypertension in Ghana.

Reviewer #3: GENERAL COMMENTS:

While it is known that NCD prevalence is rising at an alarming rate in sub-Saharan Africa, not much is known of the drivers of this and mechanisms that support sufficient NCD management in these settings. Papers that highlight the current state and discuss opportunities for proven effective interventions are surely needed. The authors used a cross-sectional design, purposively sampling health facilities, providers and patients in Ghana to describe determinants of blood pressure among people receiving treatment for hypertension. They offered no convincing argument to why they sampled purposively (given that hypertension is a common illness) or why they arrived at their sample size. No report is given of how participants were selected at sites once they met eligibility criteria. The study design precludes us from drawing any conclusions from the data. Despite many statistical tests, they are unable to show any predictors that are both meaningful and statistically significant.

SPECIFIC COMMENTS:

BACKGROUND:

1. Lines 50-52, are 21% of all patients with hypertension on treatment, or is the denominator here limited to those already aware of their hypertension status?

2. Lines 52-53. It might be helpful to consider framing this around the care cascade – interventions to improve BP control start at preventing increased blood pressure, improving screening and diagnosis and, perhaps most relevant to the context of this paper, improve treatment and control.

3. Lines 56-58: While this is true, it lays the blame for complications due to NCDs at the feet of the health system, rather than mention the effects of westernized diet and lifestyle and how corporations lobby against policies and laws that could improve health outcomes at a population level.

4. Line 61: “effective strategies are needed to respond to the rising burden of hypertension”. Agree 100%. However, this creates the e

---

## [Decision Letter · Decision Letter 1]

12 Dec 2023

PGPH-D-23-01066R1

Exploring patient-, provider-, and health facility-level determinants of blood pressure among patients with hypertension: A multi-center study in Ghana

Dear Dr. Byiringiro,

Thank you for submitting your manuscript to PLOS Global Public Health. After careful consideration, we feel that it has merit but does not fully meet PLOS Global Public Health’s publication criteria as it currently stands. Therefore, we invite you to submit a revised version of the manuscript that addresses the points raised during the review process.

We look forward to receiving your revised manuscript.

Kind regards,

Nasheeta Peer

Academic Editor

Journal Requirements:

a. State what role the funders took in the study. If the funders had no role in your study, please state: “The funders had no role in study design, data collection and analysis, decision to publish, or preparation of the manuscript.”

b. If any authors received a salary from any of your funders, please state which authors and which funders.

Additional Editor Comments (if provided):

Line 49: ‘…leading causes of death, cardiovascular disease, and stroke’. Please explain clearly Is CVD not the cause of death being referred to and does stroke not fall under CVD? This is the impression created by this text.

Lines 56-58: Please improve the sentence construction.

Line 59 What ‘public health system initiatives’ are currently in place? Or are you referring to generic efforts?

Line 71-72: Improve syntax of the entire sentence e.g., ‘…beneficial to explore factors not only affecting hypertension control…’

Lines 75-77: What about high caloric intake and obesity?

Line 90: replace ‘grew up’ with ‘increased’.

Line 162: Please include the HB-HBP scale as a supplementary table in this paper.

Lines 185-186: Please include the ‘23-item questionnaire’ as a supplementary table as well.

Line 228: Correct spelling of ‘routinely’.

Table 1:

Improve syntax – it should read ‘duration of hypertension diagnoses’.

p-value should be written and reported as <0.001 and not 0.000; please correct.

Line 340: replace ‘exams’ with tests.

Table 5:

‘Length of diagnosis’ – include the condition.

Please amend to ‘Compliance with high blood pressure therapy’.

Line 370: In line with how you’ve presented the data, amend the text ‘…were two times less likely…’ to ‘were half as likely to…’

Line 376: remove ‘which is impressive’ – this is best discussed in the Discussion section with reasons provided as to why the authors consider this ‘impressive’.

This does not correspond with the confidence in diagnosing and treating hypertension. Please comment in the Discussion.

Line 412: replace ‘oddly’ with ‘unexpectedly’. There are likely various reasons for this finding which does not make it ‘odd’.

Line 425: include ‘one’ BP measurement equipment. Note, that a single working BP machine is sub-optimal.

Line 439: replace ‘this’ with ‘latter study’ to prevent confusion.

Lines 437-449: briefly summarise hypertension control in other studies rather than listing each one individually.

Lines 450-455: The flow is poor and confusing – lower SBP in your is followed by a study from Accra and back to your study with no significant associations found. Please improve for clarity.

Line 455-464. This text is rambling and without a clear message. Please summarise succinctly your message about education, wealth and BP control with direct comparison and relevance to your study findings. Please shorten.

Lines 466-489: This paragraph is a page long, rambling and unfocused. It discusses distance travelled to clinics, strengthening of healthcare services, waiting times, etc. as well as an example from Rwanda where removing distance barriers made no impact on BP control, without discussing this contrasting finding. It should be one paragraph, one idea. What is the key message in this paragraph? Please summarise clearly and succinctly. Please shorten.

Lines 489 onwards: To avoid repeating myself, I suggest that the authors review the rest of their Discussion in line with the recommendations above. Note, this suggestion was made in the previous review as well but has not been adequately addressed. I suggest that an experienced science writer/researcher review the text before resubmitting again, please.

Reviewers' comments:

Reviewer's Responses to Questions

**Comments to the Author**

1. If the authors have adequately addressed your comments raised in a previous round of review and you feel that this manuscript is now acceptable for publication, you may indicate that here to bypass the “Comments to the Author” section, enter your conflict of interest statement in the “Confidential to Editor” section, and submit your "Accept" recommendation.

Reviewer #2: All comments have been addressed

Reviewer #3: (No Response)

Reviewer #4: All comments have been addressed

2. Does this manuscript meet PLOS Global Public Health’s publication criteria? Is the manuscript technically sound, and do the data support the conclusions? The manuscript must describe methodologically and ethically rigorous research with conclusions that are appropriately drawn based on the data presented.

Reviewer #2: Yes

Reviewer #3: No

Reviewer #4: Partly

3. Has the statistical analysis been performed appropriately and rigorously?

Reviewer #2: Yes

Reviewer #3: No

Reviewer #4: Yes

4. Have the authors made all data underlying the findings in their manuscript fully available (please refer to the Data Availability Statement at the start of the manuscript PDF file)?

Reviewer #2: Yes

Reviewer #3: Yes

Reviewer #4: Yes

5. Is the manuscript presented in an intelligible fashion and written in standard English?

Reviewer #2: Yes

Reviewer #3: Yes

Reviewer #4: Yes

6. Review Comments to the Author

Reviewer #2: Thank you for the revision and great overview of the need to address HTN in LMICs. A few thoughts on this, beta blockers as listed are not considered as best first line antihypertensive treatment for hypertension unless there was combination therapy or specific indications. Also, in regard to the affluent having increased BP, is there a correlation to their outcomes in regards to STEMIs, CV disease overall? In regards to outcomes as they have been written, what cardiovascular outcomes you referring to? Stroke and MI? Also, for the conclusion, may want to address the need for specific partnerships to address HTN and incorporate it into SDG goals as well (noncommunicable diseases).

Reviewer #3: 1. Despite concerns voiced by multiple reviewers in the previous round of review, the main statistical analysis has remained unchanged as far as I can discern. No crude estimates are provided in the main paper and it is not clear how the authors determined which variables to include in the adjusted analysis. The authors also draw inappropriate conclusions from the regression, given limitations in study methodology and overstate their findings. While they have included how they completed their sample size calculation in this version, they still do not cite or provide a reason why they used the expected effect sizes they did (lines 281-288). This is especially relevant as these differ from what they found in their study.

2. Saying that increased provider training is bad for patient outcomes is a big statement. I do not think the study methodology and results support it. Given the number of variables the authors included in their analysis, they were likely to find something that is statistically relevant. That does not in and of itself make it true or meaningful. The variable itself is problematic – if most of the training is from onboarding, providers with the least experience fall into this category. I therefore do not think it is an adequate measure of knowledge or experience and use of it introduces bias. While the authors mention this briefly in their discussion, I don’t think enough justification is provided to include it.

3. Multiple reviewers raised concerns on the length and focus of the discussion. It has been edited in some parts, but it still spans 10 pages and still lacks focus.

4. While it is the authors’ prerogative to decide how they group drugs provided in the management of hypertension, what they are listing are not antihypertensive medications as mentioned, but classes of drugs. To be using the correct terminology, they either need to refer to them as classes of drugs,or list the antihypertensives separately.

Reviewer #4: The authors have fully addressed my comments. I am happy with the revised version, although some minor revision needed mainly in terms of the language, but I will leave this to the Editor.

7. PLOS authors have the option to publish the peer review history of their article (what does this mean?). If published, this will include your full peer review and any attached files.

**Do you want your identity to be public for this peer review?** For information about this choice, including consent withdrawal, please see our Privacy Policy.

Reviewer #2: No

Reviewer #3: No

Reviewer #4: **Yes: **Kim Anh Nguyen

---

## [Decision Letter · Decision Letter 2]

13 Mar 2024

PGPH-D-23-01066R2

Exploring Patient-, Provider-, and Health Facility-Level Determinants of Blood Pressure Among Patients with Hypertension: A Multicenter Study in Ghana

Dear Dr. Byiringiro,

Thank you for submitting your manuscript to PLOS Global Public Health. After careful consideration, we feel that it has merit but does not fully meet PLOS Global Public Health’s publication criteria as it currently stands. Therefore, we invite you to submit a revised version of the manuscript that addresses the points raised during the review process.

We look forward to receiving your revised manuscript.

Kind regards,

Nasheeta Peer

Academic Editor

Journal Requirements:

Additional Editor Comments (if provided):

Overall:

Please write all text in the past tense, and not the present tense.

Please check syntax throughout the paper.

Please streamline the paper further.

Line 53: ‘the leading causes of death’ – what does this standalone phrase refer to, please? Improve sentence clarity.

Lines 515-516: Did these hospitals treat all patients with hypertension or those mainly with complicated hypertension. If the latter, then it is not comparable to hypertension control in other settings. Were these patients recruited from routine hypertension clinics?

Lines 562-567: Divide into 2 sentences and improve syntax.

Line 588: Please remove ‘the’.

Line 592: Why are this study findings different from the literature?

Line 622-624: Improve grammar: ‘results into a rush in consultation…’

Line 630: Improve syntax.

Reviewers' comments:

Reviewer's Responses to Questions

**Comments to the Author**

1. If the authors have adequately addressed your comments raised in a previous round of review and you feel that this manuscript is now acceptable for publication, you may indicate that here to bypass the “Comments to the Author” section, enter your conflict of interest statement in the “Confidential to Editor” section, and submit your "Accept" recommendation.

Reviewer #2: All comments have been addressed

Reviewer #3: (No Response)

2. Does this manuscript meet PLOS Global Public Health’s publication criteria? Is the manuscript technically sound, and do the data support the conclusions? The manuscript must describe methodologically and ethically rigorous research with conclusions that are appropriately drawn based on the data presented.

Reviewer #2: Yes

Reviewer #3: No

3. Has the statistical analysis been performed appropriately and rigorously?

Reviewer #2: Yes

Reviewer #3: No

4. Have the authors made all data underlying the findings in their manuscript fully available (please refer to the Data Availability Statement at the start of the manuscript PDF file)?

Reviewer #2: Yes

Reviewer #3: Yes

5. Is the manuscript presented in an intelligible fashion and written in standard English?

Reviewer #2: Yes

Reviewer #3: Yes

6. Review Comments to the Author

Reviewer #2: (No Response)

Reviewer #3: GENERAL COMMENTS

Thank you for the edits to this manuscript. I want to reiterate that papers that highlight the current state of hypertension management and discuss opportunities for proven effective interventions are surely needed. I still think that describing what you found at each facility and provider knowledge would be a strong paper in and of itself. Most of the remaining concerns are related to the patient-level data.

SPECIFIC COMMENTS:

1. Line 41: This does not fit with the message provided in the discussion.

2. Line 51-54: There’s jumping around between what the denominator of the reported percentage is, which makes it hard to follow. Looking at this, you’re saying only 65% are aware of their hypertension status, but 66% are on treatment. How is that possible? I suggest editing for consistency and clarity.

3. Line 55: Suggest using term “resource constrained” rather than “frail” when referring to health systems.

4. Line 160: Has this tool been validated in this setting?

5. Lines 191-198: Have you considered the impact around the “know-do” gap? It is well described in the literature that even when providers know what to do, that it doesn’t necessarily translate into practice.

6. Lines 230-231: In the response to reviewers you stated that you’ve listed these, but here you still refer to classes of drugs as first line medications.

7. Lines 235-237: Shouldn’t education be ordinal, given that it does fall into categories that can be ordered?

8. Line 247: Is it therefore fair to assume that all these variables followed a normal distribution? If not, please consider reporting medians ad IQR

9. Line 250: Again, reporting of mean implies normal distribution. Were the scores of knowledge normally distributed?

10. Lines 281-288: I would suggest that the power calculation/sample size, since it would have been done prior to participant selection, be moved to line 102.

11. Table 1: Usually, p-values are not reported in the baseline characteristics table.

12. Table 1: According to the creators of the scale, “The Hill-Bone Compliance to High Blood Pressure Therapy Scale is one of two Hill-Bone Scales. This is the original, 14-item Hill-Bone Scale developed to assess patient behaviors for three important behavioral domains of high blood pressure treatment (the three sub-scales of the original scale): Appointment Keeping (3-items), Diet (2-items), Medication Adherence (9-items).” However, you report 2 questions for appointment keeping and 3 for Diet? Can you please clarify?

13. Table 3: Technically, the urine dipstick for protein, glucose and ketones would be a single test strip. Is this not the case in Ghana?

14. Table 3 Notes: Here you are still listing classes of drug when it should be the medications.

15. Line 361-362: Does high BP therapy refer to antihypertensives? I would suggest reconsidering the phrasing here and to refrain from using “high BP therapy” as a technical term.

16. Table 5: Again, here the number of questions for different sections of the compliance score are not in line with literature describing number of questions per section.

17. Lines 376-378: CI still wide and only just significant. what happens if you stratify by urban vs rural?

18. Lines 381-382: I still don’t understand how the authors have made the determination that the mean score was “relatively good”. Relative to what? Having looked at the questions provided, they seem basic and having providers score <80% seems worrying.

19. Line 415-420: Please confirm that the values reported in text and tables match. Please also review the abstract for this.

20. Line 450-453: Please add all the numbers used in this calculation.

21. S3 Table: Please indicate which response you considered to be correct.

RESPONSES TO REVIEWERS:

22. In response to question 1 to reviewer 3, this still does not provide the information on WHY the estimate in your power calculation and what you found in your study differ so much. Which one do you think is correct? Why do they differ so much? I was not able to find this information in the methods, response or discussion.

23. While adding table 4 is useful, this does not address the underlying concern that the terminology used is not correct. You would have to refer to that list every time you mention anti-hypertensive medications. I’ve made note of instances where this is still problematic above.

7. PLOS authors have the option to publish the peer review history of their article (what does this mean?). If published, this will include your full peer review and any attached files.

**Do you want your identity to be public for this peer review?** For information about this choice, including consent withdrawal, please see our Privacy Policy.

Reviewer #2: No

Reviewer #3: No

---

## [Decision Letter · Decision Letter 3]

11 Jun 2024

PGPH-D-23-01066R3

Exploring Patient-, Provider-, and Health Facility-Level Determinants of Blood Pressure Among Patients with Hypertension: A Multicenter Study in Ghana

Dear Dr. Byiringiro,

Thank you for submitting your manuscript to PLOS Global Public Health. After careful consideration, we feel that it has merit but does not fully meet PLOS Global Public Health’s publication criteria as it currently stands. Therefore, we invite you to submit a revised version of the manuscript that addresses the points raised during the review process.

We look forward to receiving your revised manuscript.

Kind regards,

Nasheeta Peer

Academic Editor

Journal Requirements:

Additional Editor Comments (if provided):

Reviewers' comments:

Reviewer's Responses to Questions

**Comments to the Author**

1. If the authors have adequately addressed your comments raised in a previous round of review and you feel that this manuscript is now acceptable for publication, you may indicate that here to bypass the “Comments to the Author” section, enter your conflict of interest statement in the “Confidential to Editor” section, and submit your "Accept" recommendation.

Reviewer #3: (No Response)

Reviewer #4: All comments have been addressed

2. Does this manuscript meet PLOS Global Public Health’s publication criteria? Is the manuscript technically sound, and do the data support the conclusions? The manuscript must describe methodologically and ethically rigorous research with conclusions that are appropriately drawn based on the data presented.

Reviewer #3: No

Reviewer #4: Yes

3. Has the statistical analysis been performed appropriately and rigorously?

Reviewer #3: I don't know

Reviewer #4: No

4. Have the authors made all data underlying the findings in their manuscript fully available (please refer to the Data Availability Statement at the start of the manuscript PDF file)?

Reviewer #3: Yes

Reviewer #4: Yes

5. Is the manuscript presented in an intelligible fashion and written in standard English?

Reviewer #3: Yes

Reviewer #4: Yes

6. Review Comments to the Author

Reviewer #3: Thank you for your responses to my previous comments. I have no new ones.

Reviewer #4: In this revision, I have only comments regarding the presentation of Table 1.

Since the author present two subgroups of uncontrolled and controlled hypertension, it makes sense to include the column of P-values comparing the characteristics of these two.

However, with non-parametric data which the authors presented in median (IQR), the use of t-test and ANOVA test are not appropriate, and Kruskal Wallis test should be used.

7. PLOS authors have the option to publish the peer review history of their article (what does this mean?). If published, this will include your full peer review and any attached files.

**Do you want your identity to be public for this peer review?** For information about this choice, including consent withdrawal, please see our Privacy Policy.

Reviewer #3: No

Reviewer #4: No

---

## [Decision Letter · Decision Letter 4]

20 Jun 2024

Exploring Patient-, Provider-, and Health Facility-Level Determinants of Blood Pressure Among Patients with Hypertension: A Multicenter Study in Ghana

PGPH-D-23-01066R4

Dear Mr Byiringiro,

We are pleased to inform you that your manuscript 'Exploring Patient-, Provider-, and Health Facility-Level Determinants of Blood Pressure Among Patients with Hypertension: A Multicenter Study in Ghana' has been provisionally accepted for publication in PLOS Global Public Health.

Best regards,

Nasheeta Peer

Academic Editor

Please find some minor technical comments as follows:

Abstract

Line 32: Include years for the age.

Line 37-38: To prevent ambiguity, rephrase as: ‘Receiving hypertension care at government health facilities than at private health facilities was associated with…’

Methods

Line 114-115: Improve syntax – remove ‘were’ after 'facilities’.

In addition to this, we note that you have indicated that data from this study are available upon request. As you may be aware, PLOS journals require authors to make all data underlying their findings fully available without restrictions at the time of publication (https://journals.plos.org/globalpublichealth/s/data-availability). If specific legal or ethical constraints prevent public sharing of a dataset, authors must explain how others can access the data.

b) If there are no restrictions, please upload the minimal anonymized data set necessary to replicate your study findings to a stable, public repository and provide us with the relevant URLs, DOIs, or accession numbers. Please see https://www.bmj.com/content/340/bmj.c181.long for guidelines on how to de-identify and prepare clinical data for publication. For a list of recommended repositories, please see https://journals.plos.org/globalpublichealth/s/recommended-repositories. You also have the option of uploading the data as Supporting Information files, but we would recommend depositing data directly to a data repository if possible.

**Reviewer Comments (if any, and for reference):**

Reviewer's Responses to Questions

**Comments to the Author**

1. If the authors have adequately addressed your comments raised in a previous round of review and you feel that this manuscript is now acceptable for publication, you may indicate that here to bypass the “Comments to the Author” section, enter your conflict of interest statement in the “Confidential to Editor” section, and submit your "Accept" recommendation.

Reviewer #4: All comments have been addressed

2. Does this manuscript meet PLOS Global Public Health’s publication criteria? Is the manuscript technically sound, and do the data support the conclusions? The manuscript must describe methodologically and ethically rigorous research with conclusions that are appropriately drawn based on the data presented.

Reviewer #4: Yes

3. Has the statistical analysis been performed appropriately and rigorously?

Reviewer #4: Yes

4. Have the authors made all data underlying the findings in their manuscript fully available (please refer to the Data Availability Statement at the start of the manuscript PDF file)?

Reviewer #4: Yes

5. Is the manuscript presented in an intelligible fashion and written in standard English?

Reviewer #4: Yes

6. Review Comments to the Author

Reviewer #4: I have no further comment.

7. PLOS authors have the option to publish the peer review history of their article (what does this mean?). If published, this will include your full peer review and any attached files.

**Do you want your identity to be public for this peer review?** For information about this choice, including consent withdrawal, please see our Privacy Policy.

Reviewer #4: **Yes: **Kim A Nguyen
